# WasmGuard: Enhancing Web Security through Robust Raw-Binary Detection of WebAssembly Malware

## Abstract

WebAssembly (Wasm), a binary instruction format designed for efficient cross-platform execution, has rapidly become a foundational web standard, widely adopted in browsers, client-side, and server-side applications. However, its growing popularity has led to an increase in Wasm-targeted malware, including cryptojackers and obfuscated malicious scripts, which pose significant threats to web security. In spite of progress in deep learning based detection methods for Wasm malware, such as MINOS, these approaches face substantial performance degradation in adversarial environments. In our experiments, MINOS's detection accuracy dropped to 49.90% under adversarial attacks, revealing critical vulnerabilities. To address this, we introduce **WasmGuard**, a robust malware detection framework tailored for Wasm. WasmGuard employs FGSM-based adversarial training with prior-based initialization for perturbation bytes in customized sections, coupled with a novel adversarial contrastive learning objective. Using our large-scale dataset, **WasmMal-15K** (publicly available), WasmGuard outperforms six competing methods, achieving up to 99.20% Robust Accuracy and 99.93% Standard Accuracy under PGD-50 adversarial attacks, while maintaining low training overhead. Additionally, we have released **WebChecker**, a WasmGuard-powered browser plugin, providing real-time protection against malicious Wasm files.

## CCS Concepts

• **Information systems** → **World Wide Web**; • **Security and privacy** → *Web application security*.

## Keywords

Wasm malware detection, Adversarial robustness, Contrastive learning, Perturbation bytes, Web security

## 1 Introduction

WebAssembly (Wasm) is a binary instruction format and execution environment designed to provide efficient, portable, and secure execution for web-based applications. Originally created as a compilation target for high-level languages, Wasm has rapidly become a cornerstone of the modern web, enabling high-performance execution across browsers, cloud platforms, and IoT devices [29]. With support from all major browsers, it has become highly versatile for various web applications, offering near-native execution speed and cross-platform compatibility [23].

However, this widespread adoption has introduced significant security risks. A recent analysis of 12,291 Wasm samples collected between May 2018 and June 2021 by CrowdStrike found that 75% were classified as malicious [8]. These Wasm-based malware samples, often executed within web browsers, originate from languages such as C++, Rust, and JavaScript. They are commonly employed in illegal activities like cryptojacking or to conceal malicious scripts, posing a growing threat to web security [24][19].

Traditional antivirus software and browser extensions detect Wasm malware using signature-based or blacklist techniques, which can be easily circumvented by sophisticated attackers [6, 8]. Recent advancements have introduced machine learning-based detection methods, including dynamic and static approaches [13, 24]. Static detection methods, which analyze Wasm executables without executing them, provide faster detection and lower resource consumption compared to dynamic methods. Thus, this paper focuses on static detection methods. Among these, MINOS [25] represents the state-of-the-art. MINOS converts Wasm binaries into grayscale images and employs a convolutional neural network classifier, achieving an Standard Accuracy (SA) of up to 99.73% on our clean dataset. Nevertheless, such methods remain vulnerable to adversarial attacks [1, 4]. For instance, in our experiments, MINOS's Robustness Accuracy (RA) dropped sharply to 49.9% under adversarial conditions, with an Attack Success Rate (ASR) reaching 49.97%.

Current studies on Wasm malware detection have several limitations: (1) A lack of public datasets, with most studies using small, self-built datasets [1, 4, 25]; (2) The majority of machine learning-based techniques rely on feature engineering [2, 14, 15, 30], which is labor-intensive and less adaptable. Although MINOS, the state-of-the-art, uses deep learning, it still converts Wasm binaries to images rather than using raw binaries for end-to-end detection; (3) Notably, despite many studies on evasion techniques, there is still a lack of robust detection methods that can effectively defend against adversarial attacks in the evolving Wasm malware landscape.

To comprehensively address the aforementioned challenges, we propose WasmGuard, the first robust approach for detecting raw-binary Wasm malware, and construct WasmMal-15K, a large-scale dataset for Wasm malware detection research. To train a robust Wasm malware detector, WasmGuard adopts a single-step adversarial training framework based on FGSM (Fast Gradient Sign Method), a technique widely used in traditional tasks such as image recognition and malware detection for Windows and Android programs. To facilitate this adversarial training, we introduce a novel perturbation-bytes injection technique for Wasm files. Additionally, we integrate the garbage-code injection strategy to generate initial adversarial examples. Critically, we take the following measures to enhance adversarial training: (1) Prior-based initialization for perturbation-bytes to strengthen the initial adversarial examples' attack capability in single steps, and (2) Contrastive learning involving both clean and adversarial examples to optimize their representation space. As our experiments show, WasmGuard achieves substantially better robustness with RA up to 99.2% and ASR as low as 0.73%, while retaining higher effectiveness with SA up to 99.93%, compared to six competing methods, all at the cost of limited training time.

In summary, this paper contributes the following:

- We propose WasmGuard: the first robust raw-binary Wasm malware detection approach, utilizing FGSM-based adversarial training enhanced with prior-based perturbation initialization and contrastive learning involving both adversarial and clean examples. It also introduces a novel adversarial sample generation method, injecting perturbation bytes into 14 newly added custom sections, along with garbage code injection.
- We introduce WasmMal-15K[1]: a new large-scale dataset consisting of 7512 malicious and 7512 benign Wasm samples, serving as a valuable resource for research in Wasm malware detection.
- We conduct extensive experiments: to demonstrate that WasmGuard has significantly superior robustness under high-effort adversarial attacks and higher performance under clean conditions, both compared to six competing methods.
- We develop and release WebChecker[2]: a novel browser plugin using the WasmGuard technique to provide real-time alerts for malicious Wasm files on webpages.

The rest of this paper is organized as follows: Section 2 outlines the related work. Section 3 details our WasmGuard approach. In Section 4, we discuss the experiments and results. Finally, Section 6 concludes the paper.

## 2 Background

### 2.1 Wasm Malware Detection Techniques

Most existing Wasm malware detection techniques employ dynamic analysis methods, collecting runtime information such as CPU cache events [17], instruction and control flow features [2], [30], and memory and network characteristics [14]. However, these methods increase computational overhead and resource consumption, affecting user experience, and can be bypassed by attackers detecting the analysis environments. Static methods, which do not require program execution, can overcome these drawbacks. Recently, an effective static Wasm malware detection approach, MINOS, has emerged. MINOS converts each Wasm executable into a 100×100 grayscale image, and utilizes a convolutional neural network to create the detection model, achieving state-of-the-art detection performance under clean conditions. However, none of the above approaches produce adversarially robust detection model.

### 2.2 Gradient-based Generation of Adversarial Malware

There are numerous gradient-based techniques for generating adversarial examples for traditional PE (Portable Executable) malware [20, 22]. One efficient and widely used technique is the Fast Gradient Sign Method (FGSM), which perturbs the input data in the direction of the loss function gradient to generate PE adversarial examples [9]. For example, Suciu et al. [31] apply FGSM to craft perturbation bytes by injecting the adversarial payload into the slack space between sections and at the end of the PE file. However, these byte-based FGSM methods for PE malware cannot be directly

applied to Wasm malware due to their distinct file formats. To generate adversarial examples for Wasm malware, Madvex [21] proposes an image-based gradient method using code-grayscale-images and performs semantic-preserving transformations on instruction constants. Nevertheless, this method requires maintaining the mapping between code-image pixels and code binaries, resulting in high computational complexity and additional runtime overhead.

### 2.3 Wasm Binary Rewriting

Binary rewriting techniques generate functionally equivalent variants of binary code by applying rewriting rules, commonly used in malware evasion, code optimization, and other binary-level transformations [12]. Current techniques for rewriting Wasm binaries include BREWasm[5], WASMixer[6], and Wasm-Mutate[3]. BREWasm[5] provides a comprehensive framework for static binary rewriting, offering fine-grained APIs to modify and re-encode Wasm objects into valid binaries. WASMixer[6] focuses on obfuscation techniques, such as memory encryption, control flow flattening, and opaque predicates, to enhance code security. Wasm-Mutate[3] is a Wasm-specific diversification engine that utilizes lazy parsing to generate diverse variants rapidly, producing thousands of efficient variants in minutes with minimal execution overhead and mitigation of side-channel attacks. In this work, we utilize Wasm-Mutate to generate Wasm malware variants and use BREWasm to inject binary bytes for rewriting the malware binaries.

### 2.4 Wasm File Structure

As shown in Fig. 1, each Wasm file consists of Magic Code, Version Number fields, and 13 distinct sections [34]. Optional Custom Sections, which always contain non-essential data like debugging information and third-party plugins, can be customized and placed between sections or as the head or tail section. The other 12 sections include: the Type Section and Function Section for defining function types and detailing function parameters; the Import Section and Export Section for listing required module imports and accessible module exports; the Table Section and Memory Section for defining tables and memory types; the Global Section for global variables; the Start Section for initializing the module state; the Element Section for table subranges; the Code Section for containing function code; and the Data Section and Data Count Section for initializing and quantifying memory ranges.

## 3 Proposed Method

In this section, we propose our WasmGuard approach, which builds a robust and effective Wasm malware detection model using adversarial training. We first define the notations for the detection and threat model, and then detail the WasmGuard approach.

### 3.1 Notation and Threat Model

*3.1.1 Notation.* To formulate a binary Wasm malware detection task with an underlying data distribution $D$, we denote the input space and label space as $X \subset \{0, 1, ..., 255\}^*$ and $Y = \{0, 1\}$, respectively. For $\forall (x,y) \in D$, the input Wasm sample $x$ is a variable-length binary string, and the output label $y$ indicates whether it is malware. The task is to find is to find the optimal parameters $\theta$ of the classification model $F(\theta) : X \rightarrow Y$ that minimize the loss function

---

[1]https://github.com/Q8201/WasmMal
[2]https://github.com/Q8201/WasmGuard

$L(\theta, x, y)$, such as the CE (Cross Entropy) loss function, as shown in the following formula:

$$\min_{\theta} E_{(x,y) \sim D} [L(\theta, x, y)] \qquad (1)$$

*3.1.2 Threat Model.* Our WasmGuad aims to develop a robust Wasm malware detection model resilient against adversarial attacks. Adopting the standard threat modeling framework from prior work [26], we outline our assumptions about the attacker's goals, knowledge, capabilities, and methods as follows: (1) The attacker tries to cause integrity violations by inducing the detection model to misclassify malicious Wasm executables as benign and benign ones as malicious. (2) The attacker possesses complete knowledge of the malware classifier (whitebox) and can manipulate any bytes in the input binary using functionality-preserving transformations. (3) The attacker is unable to modify or influence the trained model directly, except through changes to the input. (4) The attacker can use a multi-step iterative approach, rather than a single-step one, thus enhancing the attack's intensity.

## 3.2 WasmGuard Overview

Fig. 1 provides an overview of our WasmGuard approach, illustrating the primary modules and adversarial training procedure aimed at achieving a robust and effective Wasm malware detection model. As shown in the figure, the trained detection model, based on the MalConv-GCG [27] architecture, consists of three light-blue modules: (1) a word embedding module for converting Wasm files into word embedding vectors (e.g., $x \rightarrow e$); (2) a representation module for extracting vector features (e.g., $e \rightarrow h$); (3) a classification head for producing the predicted probability output (e.g., $h \rightarrow p$).

To craft adversarial examples for adversarial training, WasmGuard introduces a novel transformation module. It generates the initial adversarial example $x^{adv}$ for each input clean sample $x$ by injecting two types of perturbations, then updates it with FGSM based on the adversarial CE loss $L_{AdvCE}$ of $x^{adv}$. Additionally, WasmGuard incorporates a projection head that consists of a fully connected layer to produce projected eigenvectors from the feature vectors (e.g., $h \rightarrow z$). It also designs an adversarial contrastive learning loss, *ACLoss*, which involves the eigenvectors of both clean and adversarial examples (e.g., $x, x^{adv}$) to optimize the representation space. During adversarial training, the *Total Loss*, which integrates *ACLoss* with the basic adversarial training loss, *ATLoss*, is propagated backward to update the detection model's parameters $\theta$ and the projection head's parameters $\theta_j$.

The adversarial training procedure of WasmGuard can be formulated as the following min-max optimization problem:

$$\min_{\theta, \theta_j} E_{(x,y) \sim D} \left[ \max_{\delta \in \xi} L_{AdvCE}(\theta, x + \delta, y) \right] \qquad (2)$$

The inner maximization seeks the optimal adversarial perturbation $\delta$ that amplifies the CE loss of the model $\theta$ when applied to the perturbed sample $x+\delta$. Here, $\xi = \{\delta : \|\delta\| \le \varepsilon\}$ represents the threat bound with the maximum perturbation strength $\varepsilon$. Conversely, the outer minimization aims to refine the model $\theta$ and the module $\theta_j$ to minimize the *Total Loss*.

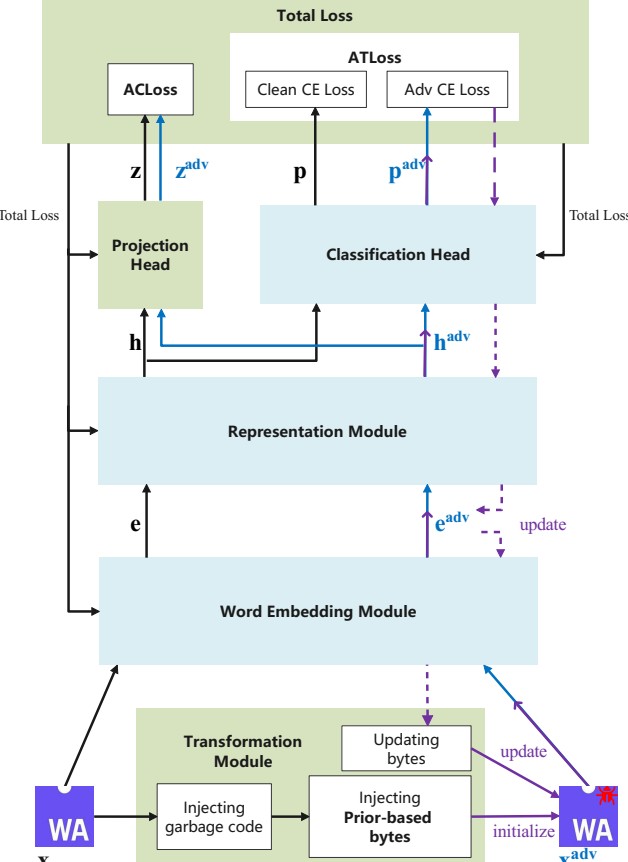

**Figure 1: Overview of WasmGuard approach.**

## 3.3 Generation of Adversarial Wasm Samples

To facilitate adversarial training, we propose an adversarial-example generation algorithm for Wasm samples, depicted in Algorithm 1. This algorithm crafts initial adversarial samples by injecting both garbage code and perturbation bytes without altering the file's functionality. As the first work to utilize adversarial bytes to perturb Wasm files, it creates 14 custom sections as injection locations shown in Fig. 2 and updates the perturbation bytes using FGSM. To implement the semantic-preserving injection of garbage code and perturbation bytes, we utilize *BREWasm* [5], a general Wasm binary rewriting tool.

Next, we will detail the three key steps of Algorithm 1:

*3.3.1 **Injecting garbage code**.* Garbage code refers to extraneous, non-functional code inserted into a program to obfuscate its structure without altering its intended behavior or violating its syntactical correctness. For each Wasm sample $x$, we insert three groups of garbage code into the file's six sections, as indicated by the striped bars in Figure 2:

- Inserting implementation of functions: First, we create $fc$ extraneous function signatures with random arguments and return values in the file's Type and Function sections. Next, in the Code section, we insert garbage instructions

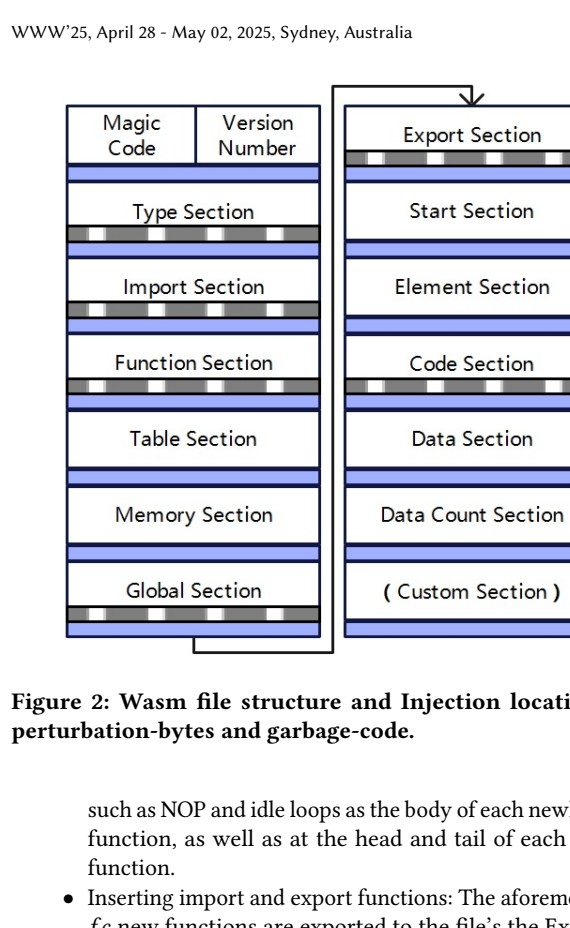

**Figure 2: Wasm file structure and Injection locations for perturbation-bytes and garbage-code.**

such as NOP and idle loops as the body of each newly-added function, as well as at the head and tail of each existing function.

- Inserting import and export functions: The aforementioned $fc$ new functions are exported to the file's the Export section. To enhance the resultant file's obfuscation, we insert the names of $im$ extraneous import functions into the Import section using the following naming strategies: For benign samples, the inserted function names are strings such as 'ransomware' and 'encrypt' to imply malicious intent; for malware, the inserted function names mimic those commonly used in benign Wasm files, such as 'clock_time_get' and 'fd_write'.

- Inserting global variables: We insert $g$ new global variables initialized with random values into the Global section. Here, $fc$, $im$ and $gl$ are adjustable hyperparameters.

*3.3.2  **Injecting initial perturbation bytes**.* For each Wasm sample, at the start of adversarial training, we create 14 new custom sections as injection locations for perturbation bytes. These sections are positioned between the existing sections, before the first section, and after the last section, as illustrated by the bluish rectangular bars in Fig. 2. During each training epoch, we insert the prior perturbation bytes from the previous training epoch into these custom sections, as lines 2-3 show in Algorithm 1. The prior perturbation bytes of the input sample, denoted by the global variable $B_p$ in this algorithm, are randomly initialized at the beginning of training and updated with the generation of the sample's perturbation bytes in each epoch.

Given *perturbation_budget*, the ratio of inserted perturbation bytes to the sample's file size, we can calculate and control the average length of the perturbation bytes injected into each custom section, *average_size*, using the following equation, where

*file_size* is the Wasm sample's file size, and *custom_num* is the number of custom sections inserted:

$$average\_size = \frac{file\_size \times perturbation\_budget}{custom\_num} \quad (3)$$

---

**Algorithm 1** Generating Adversarial Wasm Sample: $GenAdvWasm$ $(x, y, \theta, M, \varepsilon)$.

---

**Input:** Original Wasm sample $x$ and its label $y$, detection model's parameters $\theta$, word embedding matrix $M$, maximum perturbation strength $\varepsilon$.

**Output:** Perturbed Wasm sample $x^{adv}$.

**Global:** Prior perturbation bytes $B_p$ with random initialization.

1: Inject garbage code into $x$ to get $x^{adv}$
2: Get locations $L_p$ in $x^{adv}$ for injecting perturbation bytes
3: $x^{adv}[L_p] \leftarrow B_p[L_p]$  //Inject prior perturbation bytes
4: $y^{adv} \leftarrow model.Predict(\theta, x^{adv})$
5: **if** $y^{adv} \neq y$ **then**
6:   **return** $x^{adv}$.
7: **end if**
  /* Update perturbation bytes: 8-12 */
8: $e^{adv} \leftarrow model.Embed(\theta, x^{adv})$
9: $p^{adv} \leftarrow model.Classification(\theta, e^{adv})$
10: $gradient \leftarrow sign(\nabla_{e^{adv}}CrossEntropy(p^{adv}, y))$
11: Update $e^{adv}[L_p]$ using $\varepsilon * gradient[L_p]$
12: Reconstruct $x^{adv}$ using $e^{adv}$ via Eq. (4)
13: $B_p[L_p] \leftarrow x^{adv}[L_p]$  //Set prior perturbation bytes
14: **return** $x^{adv}$.

---

*3.3.3  **Updating perturbation bytes:*** After obtaining the initial adversarial sample $x^{adv}$ using the two types of injections, we update its perturbation bytes using FGSM. This updating procedure, illustrated by the purple lines in Fig. 1 and lines 8-12 in Algorithm 1, involves the following steps: First, $x^{adv}$ is sent to the word embedding module to obtain its embedding vector $e^{adv}$, which is then forward propagated to produce the predicted probability output $p^{adv}$. Next, the cross-entropy loss between $p^{adv}$ and the label $y$ is calculated. Subsequently, backpropagation is used to differentiate the word embedding vector, yielding the corresponding gradient. Finally, the gradient information is added as noise to the perturbed counterpart in $e^{adv}$, and it is reconstructed into the sample $x^{adv}$ using Eq. (4). Here, $L_p$ denotes the offset addresses in the $x^{adv}$ file for injecting perturbation bytes, $M_j$ represents the $j$th row of the word embedding matrix.

$$x^{adv}[L_p] = argmin_{j \in 0 \cdots 255}(\|e^{adv}[L_p] - M_j\|_2) \quad (4)$$

### 3.4  Loss Function

During adversarial training, to optimize the representation space of both clean and adversarial Wasm samples, WasmGuard employs a novel adversarial contrastive loss (*ACLoss*) to regulate the basic adversarial training loss (*ATLoss*). The total loss is calculated as shown in Eq. (5), where $\lambda \in [0, 1]$ is the weight factor for the regularization term.

$$Total\ Loss = ATLoss + \lambda * ACLoss \quad (5)$$

*3.4.1 **Basic adversarial training loss (ATLoss)**.* To balance robustness and accuracy, both the cross-entropy loss of the clean sample $x$ and that of the adversarial sample $x^{adv}$ are included in the basic adversarial training loss, calculated as shown in Eq. (6). Here, $N$ denotes the number of samples in a batch, $y_i$ denotes the label of sample $x_i$, and $p_i$ and $p^{adv}$ denote the probabilities that clean sample $x_i$ and adversarial sample $x^{adv}$ are predicted to be $y_i$, respectively. By minimizing the *ATLoss*, the model learns from both clean and adversarial samples, helping to classify them correctly within the decision boundary.

$$ATLoss = L_{AdvCE} + L_{CleanCE}$$

$$= -\frac{1}{N} \sum_{i=1}^{N} (y_i \log(p_i \cdot p_i^{adv}) + (1 - y_i) \log((1 - p_i)(1 - p_i^{adv}))) \quad (6)$$

*3.4.2 **Adversarial contrastive loss (ACLoss)**.* For each input batch of $N$ clean Wasm samples, WasmGuard crafts $N$ adversarial Wasm samples as described in the previous subsection. Next, WasmGuard send both clean samples (e.g., $x$) and adversarial samples (e.g., $x^{adv}$) sequentially through the word embedding module, the representation module, and the projection head. This process produces a representation batch of $2N$ normalized representation vectors of Wasm samples, comprising $N$ clean sample vectors (e.g., $z$) and $N$ adversarial sample vectors (e.g., $z^{adv}$), as illustrated in Figure 1. Because each adversarial sample is derived through a semantic-preserving injection into a clean sample, its label remains the same as that of the original clean sample. To optimize the Wasm sample representation space, we introduce supervised adversarial contrastive learning, leveraging both clean and adversarial samples with their labels, and design the contrastive loss function as follows:

$$ACLoss = \frac{1}{2N} \sum_{i=1}^{2N} \frac{-1}{|PC(i)| + |PA(i)|} L_i, \qquad where$$

$$L_i = \sum_{p \in PC(i) \cup PA(i)} \log \frac{e^{(z_i \cdot z_p / \tau)}}{e^{(z_i \cdot z_p / \tau)} + \sum_{j \in NC(i) \cup NA(i)} e^{(z_i \cdot z_j / \tau)}} \quad (7)$$

Here, for each anchor sample vector $z_i$ in a representation batch: (1) Positive samples are all samples with the same label in the batch. These consist of consists of two subsets: $PC(i)$, including all clean samples with the same label except $z_i$, and $PA(i)$, including all adversarial samples with the same label. (2) Negative samples are all samples with different labels in the batch. These consists of two subsets: $NC(i)$ and $NA(i)$, including all clean and adversarial samples with different labels, respectively.

Unlike existing supervised contrastive learning methods like SupCon [16, 36], which only use label information from clean samples, our adversarial contrastive learning approach introduces adversarial samples and leverages labels from both clean and adversarial samples. This inclusion provides a greater number and variety of harder positives and negatives, benefiting the model's generalization, robustness, and decision boundary clarity. Additionally, for each anchor-positive pair (e.g. $z_i$-$z_p$), the denominator in $L_i$ of Eq. (7), only includes the numerator, rather than including terms from other positive samples, as a normalization term to alleviate intra-class repulsion.

# 4 Experiments

## 4.1 Experimental Setup

*4.1.1 **Dataset**.* Lacking public datasets for Wasm malware detection, we construct and release a large-scale dataset called WasmMal-15K. Initially, we collected 8631 Wasm binary files from a GitHub repository [11], followed by filtering out duplicates and non-compliant files. Using VirusTotal [33] , we labeled the remaining files, obtaining 7512 benign and 62 malicious samples. To balance the dataset, we employed the mutation tool *wasm-mutate* [3] to generate 7450 functionality-preserving variants of the 62 malicious samples. This resulted in the WasmMal-15K dataset, comprising 7512 benign and 7512 malicious Wasm files. In our experiments, the dataset was split into training and testing sets in a ratio of 8 : 2.

*4.1.2 **Evaluation metrics**.* In our experiments, we evaluate each Wasm malware detection model's effectiveness under non-adversarial conditions and robustness under adversarial conditions. We assess model effectiveness on original clean samples using three standard classification metrics: SA (Standard Accuracy), FNR (False Negative Rate), and FPR (False Positive Rate). For instance, SA denotes the effectiveness on all test samples without attacks, calculated as

$$SA = \frac{Number\ of\ correctly\ predicted\ clean\ samples}{Total\ number\ of\ clean\ samples} \quad (8)$$

We evaluate model robustness on generated adversarial samples using four metrics: RA (Robust Accuracy), ASR (Attack Success Rate), R-FNR (Robust False Negative Rate), and R-FPR (Robust False Positive Rate). Here, RA and ASR are widely-used robustness metrics, with RA measuring accuracy under attacks and ASR indicating the proportion of samples misclassified after attacks, calculated as

$$RA = \frac{Number\ of\ correctly\ predicted\ adversarial\ samples}{Total\ number\ of\ adversarial\ samples} \quad (9)$$

$$ASR = \frac{Number\ of\ samples\ misclassified\ after\ attack}{Number\ of\ correctly\ classified\ samples\ before\ attack} \quad (10)$$

R-FNR is the proportion of adversarial malicious samples that are incorrectly classified as benign, while R-FPR is the proportion of adversarial benign samples that are incorrectly classified as malicious.

*4.1.3 **Compared methods**.* As the first column of Table 1 shows, we compare WasmGuard with six advanced binary-based Wasm malware detection methods. (1) MINOS [25]: a state-of-the-art Wasm malware detection technique using grayscale images from binaries as input. (2) MalConv and (3) AvastNet: two Wasm malware detection methods implemented via transfer learning from the state-of-the-art MalConv-GCG [27] and the well-known AvastNet [18] PE models. (4) MalConv+SupCon: the MalConv-GCG model enhanced with the SupCon contrastive learning loss [16]. (5) slack-FGSM: an adversarial-training-based method implemented via transfer learning from the robust slack-FGSM [31] technique for PE files, combining its adversarial training applied to MalConv with our Wasm-specific perturbation-bytes instrumentation. (6) FGSM-RS: an adversarial-training-based method implemented via transfer learning from the robust FGSM-RS [35] technique for images, integrating its adversarial training applied to MalConv with our Wasm-specific adversarial example generation algorithm.

**Table 1: Detection performance (%) of WasmGuard and competitors, 'TL' denoting 'Transfer Learning', 'Clean Test' for 'Test on clean examples without attacks', 'Adv. Test' for 'Test under adversarial attacks', 'Adv. Training' for 'Adersarial Training'.**

| Method | Clean Test | | | Adv. Test | | | | Adv. Training |
|---|---|---|---|---|---|---|---|---|
| | SA | FNR | FPR | RA | R-FNR | R-FPR | ASR | Time |
| MINOS | 99.73 | 0.33 | 0.20 | 49.90 | 50.08 | 50.15 | 49.97 | *N/A* |
| AvastNet *(TL)* | 99.83 | 0.20 | 0.13 | 37.89 | 59.46 | 66.82 | 62.05 | *N/A* |
| MalConv *(TL)* | 99.87 | **0.07** | 0.20 | 54.12 | 46.63 | 44.69 | 45.80 | *N/A* |
| MalConv+SupCon *(TL)* | 98.57 | 2.79 | 0.08 | 53.95 | 21.16 | 58.95 | 39.18 | *N/A* |
| slack-FGSM *(TL+Ours)* | 99.83 | 0.20 | 0.13 | 65.37 | 37.24 | 30.69 | 34.52 | 1076 min |
| FGSM-RS *(TL+Ours)* | 99.77 | 0.27 | 0.20 | 69.69 | 31.48 | 28.98 | 30.14 | 1089 min |
| **WasmGuard *(Ours)*** | **99.93** | **0.07** | **0.07** | **99.20** | **1.25** | **0.34** | **0.73** | **930 min** |

**Table 2: Ablation study on detection performance (%), 'w/o' denoting 'without', 'Prior init.' for 'Prior-based initialization'.**

| Method | Clean Test | | | Adv. Test | | | |
|---|---|---|---|---|---|---|---|
| | SA | FNR | FPR | RA | R-FNR | R-FPR | ASR |
| WasmGuard *(Ours)* | 99.93 | 0.07 | 0.07 | 99.20 | 1.25 | 0.34 | 0.73 |
| **w/o ACLoss** | 99.83 | 0.20 | 0.13 | 97.27 | 3.77 | 1.63 | 2.57 |
| **w/o Prior init.** | 99.83 | 0.20 | 0.13 | 96.47 | 4.19 | 2.84 | 3.37 |
| **w/o both** | 99.77 | 0.27 | 0.20 | 69.69 | 33.54 | 25.50 | 30.14 |

*4.1.4* ***Implementation detail****.* We implemented our models using Pytorch-Lightning 1.5.10 and BREWasm 1.0.5 [5], and ran all models on a GPU server with 2*RTX 4090 (24GB) cards and Ubuntu 18.04.6 LTS. In all experiments, each detection model is trained for 50 epochs with a batch size of 32, using the Adam optimizer with a learning rate of 0.0001. The performance results are averaged over three test runs. For our WasmGuard model, the loss weight $\lambda$ is set to 0.3, which is the best choice obtained after many experiments. The hyperparameters *fc*, *im*, and *gl*, which control the injection process of garbage code are all set to 100. During adversarial training, each model undergoes a low-effort FGSM attack with a 10% *perturbation_budget*. For adversarial robustness testing, each model faces a high-effort PGD-50 attack with a 20% *perturbation_budget*.

## 4.2 Performance Evaluation Without Attack

We demonstrate the detection performance of WasmGuard and compared models in Table I. The columns of 'Clean Test' show the natural performance of each model on the clean test set without attacks. Among all these models, our WasmGuard achieves the highest performance in terms of SA, FNR, and FPR. Although the state-of-the-art gray-image-based MINOS exhibits high standard accuracy, five other raw-binary-based models outperform it, possibly due to retaining longer Wasm binaries. Compared to the adversarially-trained models slack-FGSM and FGSM-RS, the non-adversarially-trained MalConv model shows better performance on the clean test. This is because the adversarial training process, while improving robustness, can sometimes compromise the natural performance on clean samples, as reported in existing studies

[7, 28]. However, our WasmGuard, while using adversarial training to enhance robustness, also manages to improve the natural performance of the model under non-adversarial conditions.

The column of 'Time' shows the adversarial training time of the three adversarially-trained models. It reveals that our WasmGuard requires the least training time. This is because, unlike the other two models that use traditional randomly initialized perturbations, WasmGuard adopts prior-guided initialization of perturbation bytes, which accelerates the adversarial training process.

## 4.3 Performance Evaluation Under Attack

We present the detection robustness of WasmGuard and six competing models in Table I. The columns of 'Adv. Test' illustrate the robustness results of each model on the adversarial test set under PGD-50 attacks with a 20% perturbation budget. As seen from these columns, when facing attacks, our WasmGuard significantly outperforms all the competing models in all robustness metrics, including RA, R-FNR, R-FPR, and ASR. Notably, the performance of the four non-adversarially-trained models drops significantly. For instance, MINOS's accuracy drops by approximately half, with FNR and FPR rising almost 50 times, and ASR reaching up to 49.97%. Compared to these models, the two adversarially-trained models (Slack-FGSM and FGSM-RS) show some improvement but still have considerable room for enhancement, with SA below 70%, and ASR, R-FNR, and R-FPR around 30%. In contrast, our WasmGuard demonstrates outstanding robustness, achieving up to 99.2% SA, ASR as low as 0.73%, R-FPR at only 0.34%, and R-FNR at only 1.25%.

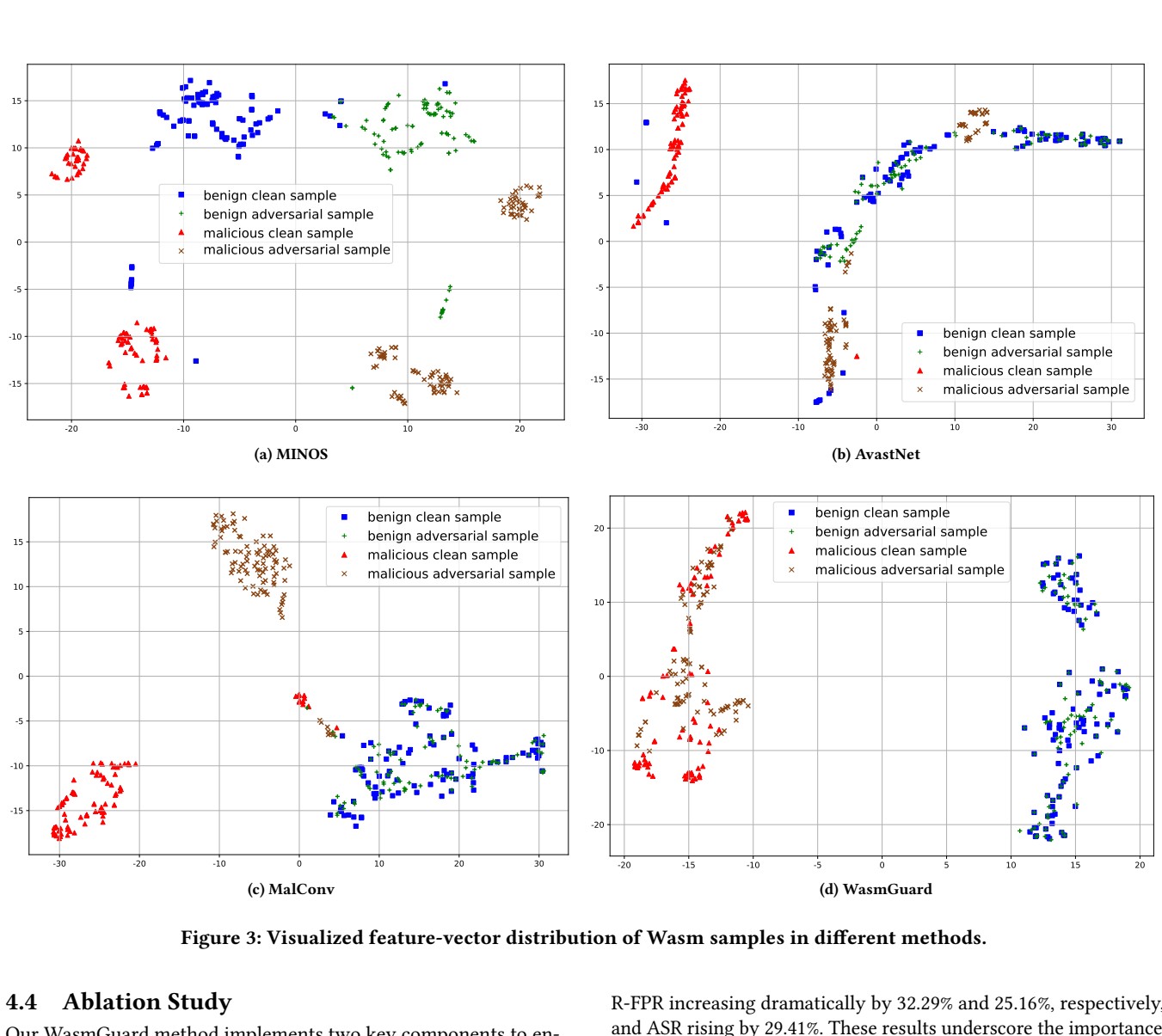

**Figure 3: Visualized feature-vector distribution of Wasm samples in different methods.**

## 4.4 Ablation Study

Our WasmGuard method implements two key components to enhance FGSM-based adversarial training: (1) ACLoss, employing adversarial contrastive learning to optimize the representation space of Wasm examples; (2) Prior-based perturbation initialization, utilizing prior perturbation bytes to initialize adversarial examples. Ablation study results for removing these two components are shown in Table 2, under both clean and adversarial conditions.

As shown in the 'clean Test' columns, under clean conditions, removing either ACLoss or prior-based initialization results in a noticeable performance drop across all metrics. This effect is even more pronounced under adversarial conditions, as illustrated in the 'Adv. Test' columns. Thus, both ACloss and Prior-based initialization are crucial for maintaining high detection effectiveness and robustness. The most significant degradation occurs when both ACloss and Prior-based initialization are removed. In this case, SA drops by 0.16%, and FNR and FPR increase by 0.20% and 0.13%, respectively. More critically, RA plummets by 29.51%, with R-FNR and

R-FPR increasing dramatically by 32.29% and 25.16%, respectively, and ASR rising by 29.41%. These results underscore the importance of combining two components to achieve optimal detection results.

## 4.5 Visual Analysis

To visually compare the detection performance of WasmGuard with existing methods, we used t-SNE [32] to visualize the Wasm sample data in two dimensions. We selected 100 benign and 100 malicious samples from the Wasm-Mal15K dataset and generated corresponding adversarial samples, resulting in a total of 400 samples for visual analysis. Fig. 3a-3d show the distribution of sample feature vectors extracted by MINOS, AvastNet, MalConv, and WasmGuard, respectively. In the figures, benign clean samples are denoted as squares, malicious clean samples as triangles, benign adversarial samples as plus-signs, and malicious adversarial samples as cross-marks.

The following observations can be made from the figures: (1) The decision boundary between benign and malicious samples in Fig. 3d is significantly clearer than in Fig. 3a-3c. (2) The proximity

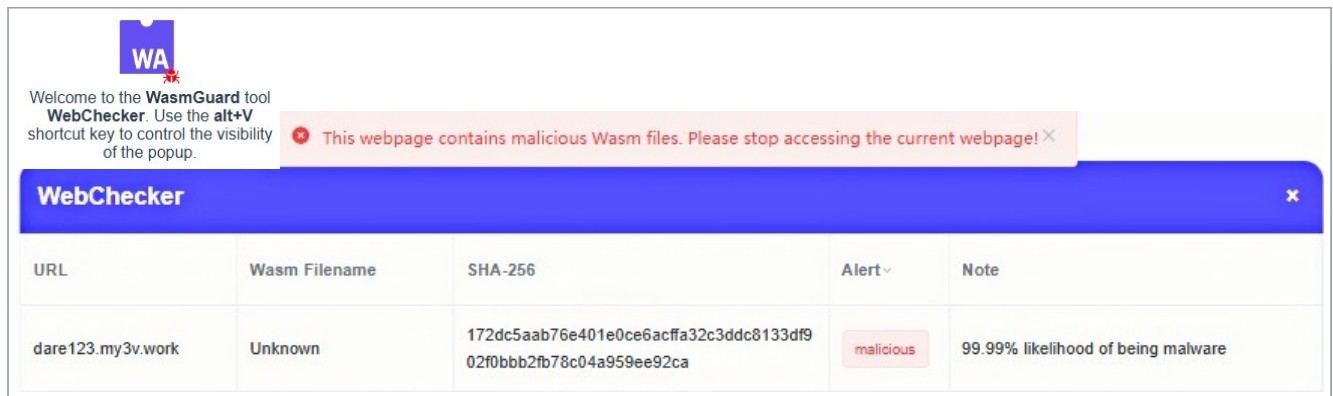

Figure 4: Example of WebChecker alert.

between benign adversarial and benign clean samples, as well as that between malicious adversarial and malicious clean samples, is much greater in Fig. 3d than in Fig. 3a-3c. Thus, it is evident that WasmGuard not only optimizes the decision boundary but also significantly improves intra-class compactness with adversarial examples involved. A clearer decision boundary leads to better detection performance, while the closer intra-class proximity between adversarial and clean samples enhances robustness. This explains why WasmGuard significantly improves detection robustness under adversarial attacks and maintains high effectiveness in the absence of attacks.

## 5 WebChecker Plugin

We developed **WebChecker**, a WasmGuard-based browser plugin designed to provide real-time alerts for webpages containing malicious Wasm files. WebChecker is built on the *vue-chrome-extension-quickstart* framework [10], an extensible base for Chrome extensions.

The plugin's technical architecture consists of a back-end and a front-end. The back-end, built using the Flask framework, is responsible for malware detection using the WasmGuard model and provides a reliable detection service interface to the front-end. The front-end collects Wasm files from webpages, displays detection results, and triggers malware alerts via a graphical user interface (GUI). As illustrated in Fig. 4, when a user accesses a webpage containing a malicious Wasm file, a popup window appears, displaying the webpage's URL, the name of the malicious Wasm file, its SHA-256 hash value, and the probability of the file being malware. The *source code, executable plugin, and detailed usage instructions* for WebChecker are accessible at https://github.com/Q8201/WasmGuard.

## 6 Conclusion

This work thoroughly investigates the robust detection of malicious WebAssembly (Wasm) binaries. We introduced WasmGuard, a resilient and efficient method for detecting WebAssembly malware, leveraging advanced adversarial example generation and adversarial contrastive learning. It integrates two types of perturbations and prior-based initialization to train a model that not only withstands strong adversarial attacks but also optimizes representation for

both clean and adversarial samples with low training overhead. We developed the WasmMal-15K dataset, which we used to thoroughly validate WasmGuard's performance against six competing methods. Furthermore, we implemented WebChecker, a WasmGuard-powered browser plugin capable of detecting Wasm malware in real time, providing practical protection for web users. The WasmMal-15K dataset and the source code of WebChecker have been made publicly available.

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
