# OpenReview forum: "WasmGuard: Enhancing Web Security through Robust Raw-Binary Detection of WebAssembly Malware"
_ACM.org/TheWebConf/2025/Conference — WWW 2025 Poster_

### Official Review · Reviewer_PHep · 2024-11-28

**Novelty:** 5
**Technical Quality:** 6

**Review:**

## summary
The growing popularity of WebAssembly (Wasm) has led to an increase in Wasm-targeted malware. Current studies on Wasm malware detection lack public datasets, rely heavily on feature engineering, and lack robust detection methods. Therefore, the authors propose WasmGuard, which employs FGSM-based adversarial training with prior-based initialization for perturbation bytes. Results show that WasmGuard achieves good performance
## Comment
First of all, this paper proposed an easy but useful model for binary Wasm malware detection. However, I would still like to make a few suggestions. Hope my opinions/suggestions will help improve your paper.
- **Lack of Explanation.** In Section 3.1, the authors mentioned the input space X. However, it is unclear why the input space is defined as ranging from 0 to 255. Is this due to 8-bit values or other reasons? It makes me confused.
- **Lack of Results on Injecting Location** In Section 3.3, the authors mentioned that they created 14 sections as injection locations. However, there are no results provided to demonstrate the influence of locations.
- **Selection of FGSM.** FGSM is an important component of the adversarial training process. However, given the numerous adversarial attack methods available (e.g., C&W), why was FGSM specifically chosen?
- **Minor Issues.**
  - In Section 2.4, the authors mentioned Fig. 1. However, the content did not match the figure.

**Questions:**

## Questions
- What is the reason for defining the input space as ranging from 0 to 255?
- What is the impact of injection locations?
- Why do you choose FGSM?

**Reviewer Confidence:**

3: The reviewer is confident but not certain that the evaluation is correct

**Scope:**

4: The work is relevant to the Web and to the track, and is of broad interest to the community

---

### Official Review · Reviewer_mW2d · 2024-12-01

**Novelty:** 5
**Technical Quality:** 6

**Review:**

This paper presents a robust and efficient framework for detecting WebAssembly (Wasm) malware. It addresses critical issues in Wasm security by introducing innovative adversarial training methods and a novel dataset for evaluation.

Strengths
1. The introduction of WasmGuard with FGSM-based adversarial training and adversarial contrastive learning significantly enhances detection robustness, achieving up to 99.2% Robust Accuracy (RA).
2. The creation of WasmMal-15K, a publicly available, large-scale dataset with equal distributions of benign and malicious Wasm binaries, fills a gap in the research community.
3. The development of WebChecker, a browser plugin powered by WasmGuard, demonstrates the practical relevance of the work.
State-of-the-Art Performance: WasmGuard consistently outperforms six baseline models (e.g., MINOS, MalConv) under adversarial attacks.

Weaknesses
1. The performance of WasmGuard in environments with advanced obfuscation techniques or novel attack methods remains underexplored.
2. The paper mainly relies on garbage code and perturbation byte injection, which may not provide sufficient complexity or diversity to cover real-world malicious software attack scenarios.
3. While the focus on static detection is justified, combining static and dynamic methods could provide a more comprehensive solution.

**Questions:**

1. How does WasmGuard perform against highly obfuscated Wasm malware, which may introduce functionality-preserving transformations?
2. Would integrating dynamic analysis features further enhance WasmGuard’s performance, especially in adversarial scenarios?
3. What measures do you propose to prevent misuse of WasmGuard for malicious purposes, such as crafting adversarial binaries?
4. During the generation of adversarial samples, have you explored other types of perturbations, such as functional transformations or more complex encoding strategies, to enhance the diversity of adversarial samples?

**Reviewer Confidence:**

3: The reviewer is confident but not certain that the evaluation is correct

**Scope:**

4: The work is relevant to the Web and to the track, and is of broad interest to the community

---

### Official Review · Reviewer_GRfx · 2024-12-01

**Novelty:** 3
**Technical Quality:** 3

**Review:**

The paper introduces WasmGuard, a novel framework for detecting WebAssembly (Wasm) malware.
WasmGuard uses adversarial training techniques, including a novel perturbation-bytes injection method and contrastive learning, to achieve high accuracy and robustness against attacks.
The authors also created a large-scale dataset, WasmMal-15K, and a browser plugin, WebChecker, to demonstrate and apply WasmGuard's capabilities.

While the topic is timely and relevant for the web, I see several problems with the current paper.
First, the motivation is not very convincing and does not match the results.
The paper never mentions what Wasm malware could even do.
Given that Wasm is limited to computation and does not have direct access to anything else, it is, by design, very limited.
While interaction is possible via JavaScript functions, they need to be specified for Wasm modules.
Looking at the cited source from CrowdStrike [8], they see 2 types of Wasm "malware":
Miners and obfuscation for JavaScript code.
Browser-based miners are no longer a relevant threat, with less than 1% of Wasm binaries being miners [A].
This leaves only obfuscation as a somewhat malicious use case.
The data gathered in the paper also reflects this change: only 62 out of the 8631 samples were malicious, i.e., <1%, which is not even close to the 75%.

Given this imbalance between malicious and benign samples, I don't think the evaluation is correct.
The 62 samples are artificially extended to 7512 samples, then an 8:2 split is used.
However, as the artificial modification does not *really* change the sample, the test and training sets are essentially the same.
For a proper evaluation, the split has to be done with the 62 samples to ensure that the underlying sample is not used for test and training.

The adversarial modifications proposed in the paper are not strong.
Inserting garbage can be easily reverted, as the inserted functions are not connected to the control-flow graph.
Thus, this does not create realistic variants of the underlying malicious samples.
There are much stronger techniques, such as self-decrypting malware and code diversification.
Such techniques have also been shown for Wasm, which are highly effective against MINOS [B].

It is further not clear why the paper mixes VirusTotal and MINOS.
Why are the samples not classified using MINOS?
If VirusTotal performs better, it should also be included in the evaluation.

While the false-positive rate sounds small, it is likely too high for real-world applications.
It would have been great to see such a usability evaluation of the browser extension.
A FPR of 0.07% still means that roughly 1 out of every 1000 scripts is flagged as malicious.
Depending on the user and how many regular websites use Wasm, this might lead to many wrong alerts.


[A] Hilbig et al. An Empirical Study of Real-World WebAssembly Binaries

[B] Cabrera-Arteaga et al. WebAssembly Diversification for Malware Evasion

**Questions:**

Why is sometimes VirusTotal and sometimes MINOS used?

Did the authors try more sophisticated obfuscation techniques, as shown by Cabrera-Arteaga et al.?

What do the malicious Wasm samples do?

**Reviewer Confidence:**

3: The reviewer is confident but not certain that the evaluation is correct

**Scope:**

3: The work is somewhat relevant to the Web and to the track, and is of narrow interest to a sub-community

---

### Official Review · Reviewer_1JFZ · 2024-12-02

**Novelty:** 5
**Technical Quality:** 5

**Review:**

## Summary
In this paper, the authors introduce WasmGuard, a framework for detecting WebAssembly malware using adversarial training and contrastive learning. WasmGuard significantly outperforms existing methods in both standard and adversarial conditions, and they also present WasmMal-15K, a large-scale dataset for Wasm malware research, along with WebChecker, a browser plugin for real-time detection.

## Strengths
The paper is well-structured and provides a comprehensive overview of the problem, related work, methodology, and results. The experiments are thorough, and the results are clearly presented with appropriate metrics.

## Weaknesses
Some sections could benefit from more detailed explanations, particularly the technical aspects of the adversarial training process and the implementation details of the WebChecker plugin. Also, there is limited discussion on real-world applications and broader impact.

**Questions:**

1. What are the computational requirements for training and deploying WasmGuard?
2. Can you add more details on the choice of hyperparameters for adversarial training and their impact on the model's performance?
3. How do you plan to maintain and update the WasmMal-15K dataset to ensure its relevance over time? Have you obtained clearance from your IRB in case you plan to release this dataset?

**Reviewer Confidence:**

1: The reviewer's evaluation is an educated guess

**Scope:**

3: The work is somewhat relevant to the Web and to the track, and is of narrow interest to a sub-community